# Discovering Mental Health Research Topics with Topic Modeling

**Xin Gao** [1]  **Cem Sazara** [1]

## Abstract

Mental health significantly influences various aspects of our daily lives, and its importance has been increasingly recognized by the research community and the general public, particularly in the wake of the COVID-19 pandemic. This heightened interest is evident in the growing number of publications dedicated to mental health in the past decade. In this study, our goal is to identify general trends in the field and pinpoint high-impact research topics by analyzing a large dataset of mental health research papers. To accomplish this, we collected abstracts from various databases and trained a customized Sentence-BERT based embedding model leveraging the BERTopic framework. Our dataset comprises 96,676 research papers pertaining to mental health, enabling us to examine the relationships between different topics using their abstracts. To evaluate the effectiveness of the model, we compared it against two other state-of-the-art methods: Top2Vec model and LDA-BERT model. The model demonstrated superior performance in metrics that measure topic diversity and coherence. To enhance our analysis, we also generated word clouds to provide a comprehensive overview of the machine learning models applied in mental health research, shedding light on commonly utilized techniques and emerging trends. Furthermore, we provide a GitHub link[*] to the dataset used in this paper, ensuring its accessibility for further research endeavors.

## 1. Introduction

The COVID-19 pandemic, which has significantly impacted our lifestyle for nearly two years, has led to a rise in psychosocial stressors and mental health problems. Consequently, there has been a notable surge in mental health research as a response to these challenges. To understand the specific topics studied by the research community, we employed topic modeling methods on the titles and abstracts of conference and journal research papers focused on mental health field. Our primary objective is to identify studies aimed at improving mental health and analyze the prominent research topics of the past decade.

To accomplish this, we collected abstracts from various databases, including arXiv, ACM, bioRxiv, medRxiv, and PubMed, spanning the period from Jan 2010 to March 2023. This extensive dataset reveals a growing interest in mental health-related research during the last decade, with a significant peak occurring during the COVID-19 pandemic. Our study aims to identify key trends and significant research topics by analyzing a dataset of 96,676 research papers. To extract meaningful insights from the dataset, we employed a Sentence-BERT based embedding model called BERTopic (Grootendorst, 2022). BERTopic generates document embeddings and clusters them into semantically coherent topics, enabling further analysis. By applying this model, we can identify specific concepts associated with each topic, providing a basis for further analysis and investigation. For instance, the topic related to suicide prominently includes terms such as "suicidal," "ideation," and "attempt." These identified topics may help uncover interdisciplinary connections and foster collaboration among different fields.

In evaluating our approach, we conducted performance evaluation using various metrics (Ferdinand Kapl, 2022). These metrics included TD (Topic Distinctiveness/Diversity), measuring topic uniqueness and diversity. Inv. RBO (Inverted Rank-Biased Overlap) evaluates topic coherence and word order similarity. NPMI (Normalized Pointwise Mutual Information) measures semantic coherence within topics by calculating the average pairwise similarity between words within a topic. Cv (Coefficient of Topic Coherence) assesses coherence among top-ranked words. Higher metric values indicate better topic coherence and interpretability. By evaluating these metrics, we optimized topic modeling by experimenting with hyperparameters. Our goal was to find the configuration that yielded topics with high coherence, diversity, and interpretability, as indicated by the metrics mentioned above. These performance evaluation

[1]Amazon Web Services, Seattle, WA, USA. Correspondence to: Xin Gao <goxi@amazon.com>.

*Workshop on Interpretable ML in Healthcare at International Conference on Machine Learning (ICML)*, Honolulu, Hawaii, USA. 2023. Copyright 2023 by the author(s).

[*]https://github.com/stella-gao/Mental-Health-Research-Paper-Dataset

metrics provided valuable insights into the quality of the generated topics and guided our decision-making process in selecting the values that optimized the topic modeling results. We evaluated the BERTopic based model's effectiveness by comparing it against two other state-of-the-art methods, namely Top2Vec (Angelov, 2020) and LDA-BERT (Basmatkar & Maurya, 2022) model. The BERTopic based model outperformed the others in metrics such as Inv. RBO and Cv. By examining the relationships between different topics using the abstracts, we gained valuable insights into the landscape of mental health research. To enhance our analysis, we designed a custom prompt to access the OpenAI API (OpenAI, 2023a) and utilized the power of the GPT-3.5-turbo model (OpenAI, 2023b) to retrieve the machine learning methods employed in the research papers. Additionally, we generated a word cloud to visually represent our findings, emphasizing the most commonly used and emerging machine learning techniques in mental health research. The resulting word cloud offers a comprehensive overview of the machine learning models applied in mental health research, shedding light on both commonly utilized techniques and emerging trends. This approach can provide valuable insights into the adoption and evolution of computational techniques in addressing mental health challenges.

## 2. Related Work

Topic modeling has been a popular method in natural language processing for some time, even preceding the advent of deep learning approaches (Wallach, 2006; Chauhan & Shah, 2021). Recently, there has been significant effort and high interest in improving mental health in the research community. Researchers have used a variety of topic modeling techniques, including Latent Dirichlet Allocation (LDA) and Top2Vec, to identify latent themes and topics in mental health content (Gaur et al., 2018; Yanchuk et al., 2022). Some studies have also explored the use of advanced natural language processing methods, such as using BERT (Bidirectional Encoder Representations from Transformers) as a text mining approach for mental health prediction (Zeberga et al., 2022). The authors employed word2vec and BERT with Bi-LSTM to effectively analyze and detect depression and anxiety signs from social media posts of Reddit and Twitter. One challenge in topic modeling for mental health research papers is ensuring that the identified topics are clinically meaningful and relevant. In addition, the quality of the data used for analysis, such as social media data, can vary widely, making it difficult to draw accurate conclusions.

Studies using topic modeling techniques have provided insights into mental health issues and identified hidden themes and patterns in mental health content. Lin et al. (Lin et al., 2022) compared different neural topic modeling methods in

learning the topical propensities of various psychiatric conditions from psychotherapy session transcripts parsed from speech recordings. Leung et al. (Leung & Khalvati, 2022) aimed to identify psychosocial stressors during the COVID-19 pandemic by applying natural language processing (NLP) to social media data and analyzing the trend in the prevalence of stressors at different stages of the pandemic. Gong et al. (Gong & Poellabauer, 2017) proposed a novel topic modeling-based approach to perform context-aware analysis of recordings, outperforming context-unaware methods and challenge baselines for all metrics. Rio-Chanona et al. (del Rio-Chanona et al., 2022) used the NRC emotion lexicon method for sentiment analysis and a structural topic model to study the causes of the 2021 Great Resignation and investigate changes in work- and quit-related posts on Reddit.

This study fills a research gap by exploring mental health research papers using topic modeling. We investigate the effectiveness of the Sentence-BERT based embedding model and compare it with other topic modeling methods. Given BERT's success in natural language processing tasks, we aim to leverage its potential for analyzing mental health content.

## 3. Topic Modeling Methods

We utilized the open-source Python library BERTopic (Grootendorst, 2022) that facilitates topic modeling using the BERT model. This method utilizes pre-trained language models to extract embeddings from text which gives it an advantage over the classical approaches such as Latent Dirichlet Allocation (LDA) (Blei et al., 2003) and Non-Negative Matrix Factorization (NMF) (Lee & Seung, 2000) that describe documents with Bag-of-Words representation. There are three main steps in this method:

EXTRACTING DOCUMENT EMBEDDINGS. This model uses a pre-trained Sentence-BERT (SBERT) (Reimers & Gurevych, 2019) model to extract document embeddings. Sentence-BERT is considered a state-of-the art method to extract document embeddings.

DIMENSIONALITY REDUCTION AND CLUSTERING. The extracted embeddings are not directly used to build the final topics. First, they go through dimensionality reduction with the Uniform Manifold Approximation and Projection (UMAP) (McInnes et al., 2018) method. UMAP method has proven to be an effective method to preserve local and global features of high dimensional data in their reduced dimensions. The embeddings after the dimension reduction are clustered with the Hierarchical Density-Based Spatial Clustering of Applications (HDBSCAN) (McInnes et al., 2017) method.

BUILDING TOPIC REPRESENTATIONS. In order to differentiate one topic from another, a modified TF-IDF procedure is used. In the regular TF-IDF representation, each word is assigned a weight that considers a local term: Term Frequency (TF) and Global term: Inverse Document Frequency (IDF). As shown in equation (1), term frequency $tf_{t,d}$ is calculated for term t and document d and inverse document frequency is calculated as the logarithm of the number of documents N in the corpus divided by the total number of documents that contain term t ($df_t$).

$$W_{t,d} = tf_{t,d} \cdot \log\left(\frac{1+N}{df_t}\right) \quad (1)$$

BERTopic, uses a modified version of TF-IDF as shown in equation (2). Term frequencies $tf_{t,c}$ are calculated by concatenating all documents in one cluster and considering it as a single cluster document c. IDF is also modified and replaced by an inverse cluster frequency term. It is calculated by taking the logarithm of the average number of words per cluster (A) divided by the frequency of term t across all clusters ($tf_t$). This term measures how much information a term provides to a specific cluster. Overall, this TF-IDF based approach generates topic-word distributions for each cluster of documents.

$$W_{t,c} = tf_{t,c} \cdot \log\left(\frac{1+A}{tf_t}\right) \quad (2)$$

The dynamic model follows a similar procedure. The main idea is that the same topics exist over different times, but they are represented differently at different times. The dynamic model extends the same TF-IDF equation to specific time steps. Term frequency is calculated using the documents available at time i and the IDF term stays the same as before. Its formula is given in equation (3). This equation allows fast calculations as the local representations can be quickly calculated without recreating the clusters and embeddings.

$$W_{t,c,i} = tf_{t,c,i} \cdot \log\left(\frac{1+A}{tf_t}\right) \quad (3)$$

We compared the BERTopic based model with two other topic modeling techniques: Top2Vec and LDA-BERT. Top2Vec is an algorithm designed for topic modeling ad semantic search tasks. It aims to automatically detect and extract topics from a given text corpus while generating jointly embedded vectors for topics and documents. Top2Vec and BERTopic share some similarities. They both use pretrained embedding models to generate word embeddings which are also then reduced by UMAP and clustered by HDBSCAN.

The difference is that Top2Vec produces clean and coherent topic representations by excluding noisy documents that are not firmly assigned to any specific topic. It effectively captures underlying themes in noisy or unstructured data. LDA-BERT is a hybrid approach for topic modeling that combines Latent Dirichlet Allocation (LDA) with Sentence BERT, an autoencoder and K-means clustering. It aims to enhance the topic discovery process and extract meaningful themes from a collection of sentences or documents, leading to improved topic coherence and interpretability.

## 4. Dataset

We compiled a comprehensive dataset of mental health-related papers by extracting them from well-known open access archives and multiple databases, namely arXiv, ACM, bioRxiv, medRxiv, and PubMed. To ensure a thorough collection, we utilized a search query specifically tailored for mental health keywords. This query, consisting of 26 relevant keywords such as Agoraphobia, Anxiety Disorder, Attention-Deficit/Hyperactivity Disorder [ADHD], Autism Spectrum Disorder [ASD], Post-Traumatic Stress Disorder [PTSD], and Schizophrenia, was constructed based on mental health statistics provided by the National Institute of Mental Health[†]. By combining these keywords using logical operators (AND and OR), we ensured the matching of relevant keywords in both the title and the abstract of research paper metadata. This approach allowed us to capture a broader range of necessary keywords for comprehensive retrieval.

The data collection process followed a systematic approach, starting with PubMed, a prominent biomedical literature database known for its extensive coverage of mental health-related research papers. Subsequently, we gathered papers from bioRxiv, medRxiv, arXiv, and ACM databases to ensure a comprehensive representation of diverse perspectives from both medical and computational domains. This sequential approach allowed us to build a robust dataset comprising 96,676 distinct research papers related to mental health.

The dataset distribution from different databases is as follows: 75,842 papers from PubMed, 12,237 papers from bioRxiv, 7,522 papers from medRxiv, 559 papers from arXiv, and 516 papers from ACM. Table 1 displays the frequency of papers published in the field of mental health from Jan 2010 to March 2023. Notably, we observed a significant growth in mental health research papers over the last decade, indicating an increasing focus on mental health in both real-world and research communities. Since 2020, a remarkable increase has been observed in multiple mental health categories, including depression, anxiety, and covid or pandemic related mental disorder. The observed surge in research and

---

[†]https://www.nimh.nih.gov/health/statistics

*Table 1.* Paper Frequency by Year in Mental Health

| Year | Frequency | Year | Frequency | Year | Frequency |
|------|-----------|------|-----------|------|-----------|
| 2023 | 413 | 2022 | 21846 | 2021 | 20663 |
| 2020 | 13814 | 2019 | 8993 | 2018 | 1688 |
| 2017 | 6257 | 2016 | 2745 | 2015 | 2196 |
| 2014 | 4595 | 2013 | 1924 | 2012 | 5339 |
| 2011 | 4758 | 2010 | 1445 | | |

*Table 2.* Prompt Provided to OpenAI gpt-3.5-turbo Model for Filtering Primary Machine Learning Models

| Role | Content |
|------|---------|
| system | "You are a helpful text summarization assistant." |
| user | "Given title and abstract of the research paper in the format [Title:Abstract], generate the machine learning model name if they used machine learning technique in the format [Model: your ML model] for the {user_text}" |

focus on mental health reflects the growing recognition of the profound impact of the pandemic on psychological well-being and the pressing need to develop effective strategies for support and intervention. These findings emphasize the critical importance of prioritizing mental health care and resilience-building efforts, both during times of crisis and beyond, to mitigate the adverse effects of such challenging circumstances.

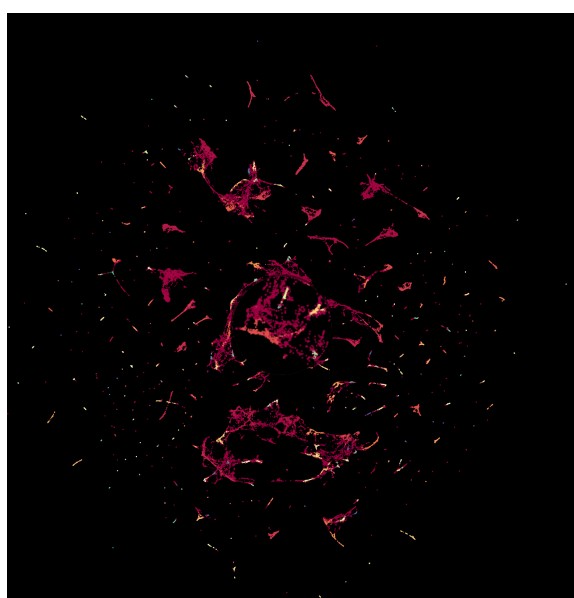

*Figure 1.* 2D UMAP Visualization of BERTopic Clusters

## 5. Methods and Results

In our study on mental health research papers, we employed the BERTopic framework to analyze the combined content of the title and abstract sections. By focusing on these key sections, we aimed to extract the primary topics and themes while minimizing noise and irrelevant details.

To ensure the integrity and consistency of our dataset, we conducted preprocessing steps such as lowercase conversion, standardizing text format, and eliminating irrelevant information like URLs, mentions, and hashtags. We retained only alphabetic characters for further analysis.

Following the data cleaning process, we applied several tokenization methods to transform the text into a sequence of tokens. We removed stop words, expanded contractions, and performed lemmatization on the remaining tokens. These preprocessing steps aimed to enhance the quality of the text data and ensure that the topic modeling method could generate precise and meaningful insights from the dataset.

We utilized the all-MiniLM-L6-v2 pre-trained embedding model from HuggingFace hub as the foundation to train the model[‡]. This embedding model is based on the latest advancements in sentence-transformers and has been trained on a vast and diverse dataset comprising over 1 billion training pairs. Its extensive training ensures that it excels as a sentence-transformers model, mapping sentences and paragraphs into a high-dimensional vector space of 384 dimensions, striking a balance between performance and efficiency.

Throughout our experimentation, we carefully identified the optimal settings that yielded the most coherent and diverse topics. To capture different levels of semantic information within the text, we considered unigrams, bigrams, and trigrams during the topic generation process. This approach allowed us to extract meaningful insights at various granularities. Additionally, we employed the HDBSCAN clustering algorithm, known for its density-based clustering capabilities, which automatically determines the number of clusters without the need for explicit specification.

To further fine-tune the model, we conducted extensive experiments with various hyperparameter values. One crucial consideration was ensuring that the generated topics were substantiated by a sufficient number of associated documents. Therefore, we set a minimum requirement of 50 documents per topic, ensuring that the topics were representative of significant patterns within the data. Similarly, we imposed a minimum threshold of 50 documents for each cluster, guaranteeing that the clusters contained substantial data for meaningful analysis and interpretation. These thresholds helped maintain the robustness and significance of the identified topics and clusters.

We performed a series of experiments by varying these hy-

---

[‡]https://huggingface.co/sentence-transformers/all-MiniLM-L6-v2

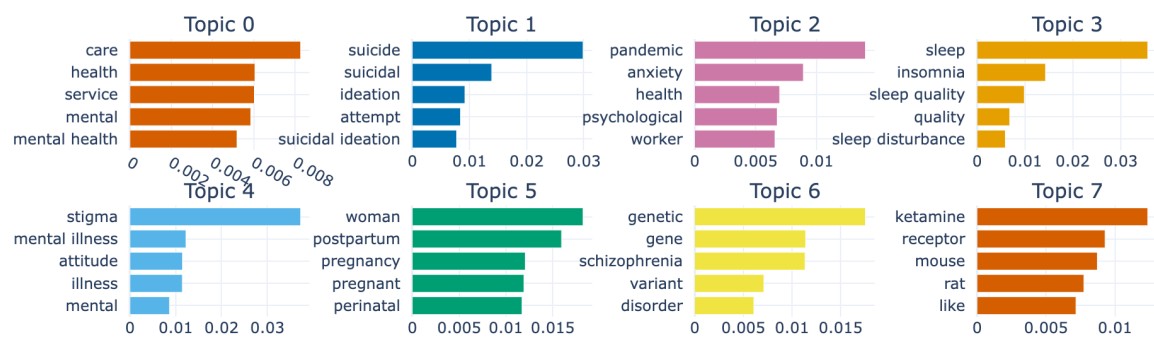

*Figure 2.* Word Scores of Top 8 Topics from BERTopic

*Table 3.* Performance Comparison of BERTopic, LDA-BERT and Top2Vec methods using four evaluation metrics

| Algorithm | No. Topics | TD | Inv. RBO | NPMI | Cv |
|---|---|---|---|---|---|
| BERTopic | 202 | **0.7208** | **0.9948** | **0.2892** | **0.8068** |
| LDA-BERT | 100 | 0.2117 | 0.7634 | 0.0772 | 0.5814 |
| Top2Vec | 134 | 0.4978 | 0.9669 | -0.0358 | 0.4654 |

perparameters and assessed the quality of the generated topics based on coherence, diversity, and meaningfulness. We evaluated the resulting topics using established metrics such as topic coherence, topic uniqueness, and semantic similarity. Starting with different values for the minimum requirement, we iteratively adjusted the threshold and analyzed the impact on the generated topics. We measured the coherence of the topics using metrics like NPMI and Cv, ensuring that the topics exhibited meaningful associations between words within each cluster. We also considered the diversity and representativeness of the topics, ensuring that they captured a significant number of documents and covered a wide range of mental health-related concepts.

Similarly, for the UMAP projection, we experimented with different values for the number of nearest neighbors. We projected the documents onto various dimensional spaces and assessed the quality of the resulting embeddings. We examined the distribution of documents in the reduced space and evaluated the preservation of both local and global structure within the data. We paid particular attention to the semantic similarity between documents, as captured by the cosine distance metric, and assessed the effectiveness of different nearest neighbor values in retaining relevant information. By systematically fine-tuning these hyperparameters and carefully evaluating the quality of the generated topics and embeddings, we identified the values of 50 associated documents per topic, 50 documents per cluster, and 20 nearest neighbors for the UMAP projection that yielded the most coherent, diverse, and meaningful results. These values were selected to ensure that the topics and clusters were robust, representative, and capable of providing comprehensive insights into the mental health research landscape.

The trained BERTopic model yielded 202 topics. These topics represent clusters of related words and phrases that frequently co-occur in the text and have been ranked based on their coherence and distinctiveness scores. Analyzing these topics provides valuable insights into the major themes and trends present in the dataset. Furthermore, utilizing BERTopic allows us to explore the relationships between different topics and visualize them.

To visualize the relationships between the generated topics, we applied UMAP once again to reduce the dimensionality of the topic embeddings to 2 and plotted them as a scatterplot. In Figure 1, we present a 2D UMAP visualization of the clusters generated by BERTopic. This visualization offers a comprehensive view of the distribution of topics in a 2D space, providing a visual summary of the relationships between different topics and shedding light on the overall structure of the dataset. The scatterplot clearly exhibits distinct clusters of topics, where topics that are closer to each other in the plot belong to the same cluster. The spread and distribution of topics in the scatterplot reflect the diversity of the mental health research landscape. The scatterplot demonstrates a wide range of themes and subtopics, with topics appearing in different regions of the plot. This diversity indicates the presence of various research directions and perspectives within the dataset. By considering the distinct clusters, topic proximity, and overall distribution of topics in the scatterplot, we can gain insights into the major themes and trends in mental health research.

Figure 2 provides valuable insights into the key themes and topics in mental health research, serving as a useful resource for identifying relevant words and topics for further analysis.

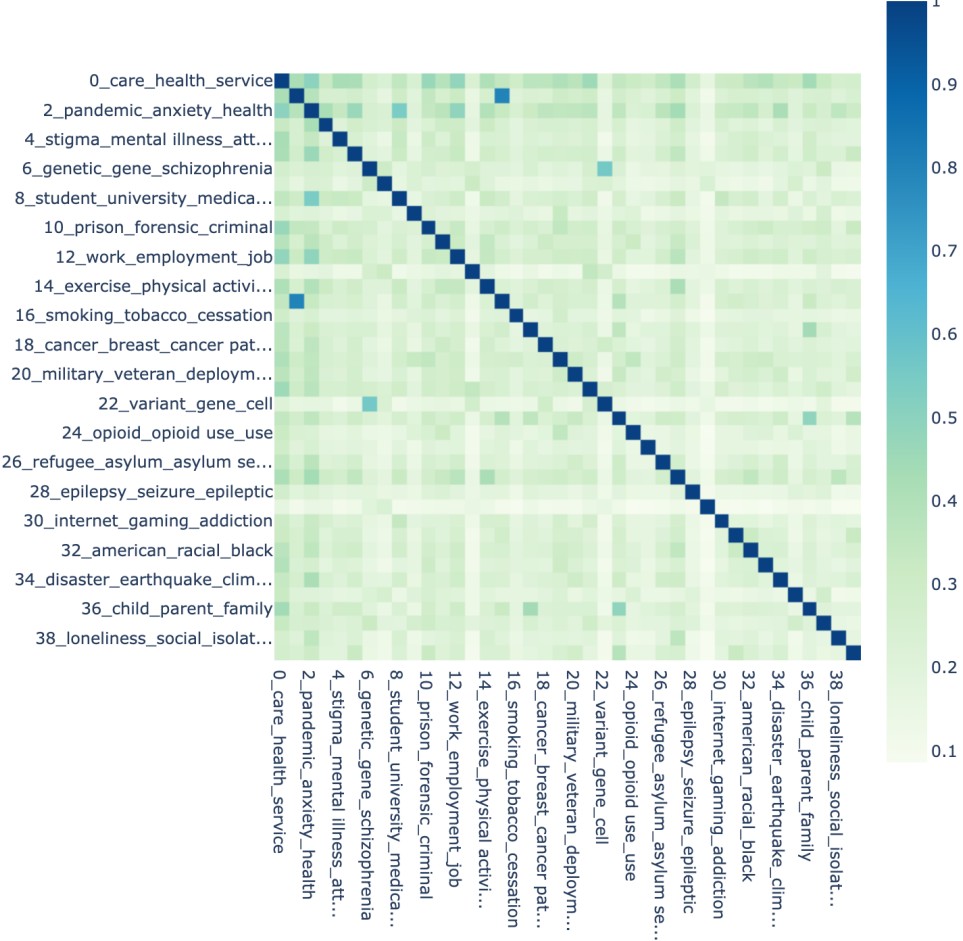

*Figure 3.* Similarity Matrix between Topics in BERTopic

The word scores associated with the top 8 mental health-related topics indicate the relevance of specific words to each topic. On the x-axis, the word scores range from 0 to 1, while the y-axis represents the topics numbered from 0 to 7. Higher word scores indicate a stronger association between the word and the corresponding topic.

Each topic is represented by a cluster of words, and the word scores indicate the degree to which each word is associated with the topic. As seen in the figure, topic 2 represents anxiety during pandemic, words such as "pandemic", "anxiety", "health", "psychological" and "worker" have high scores, indicating their strong association with this topic. Similarly, in topic 3, which is related to sleep quality and insomnia, words such as "sleep", "insomnia", "sleep quality", and "sleep disturbance" have high scores. In addition, we observed that Topic 4 was related to stigma and mental illness, while Topic 6 was related to genetic factors, such as genes, schizophrenia, variants, and disorders. These findings suggest that there are multiple factors that contribute to mental illness and schizophrenia, including both social and biological factors.

Figure 3 shows the similarity matrix between topics, which provides a valuable insight into the relationships between different topics in a given field. By analyzing the similarity matrix, we can identify clusters of topics that are closely related to each other and those that are more distant. This information can help us understand the underlying structure of the field and identify the most important topics and their interconnections. The fact that topics 6 (genetic gene schizophrenia) and 22 (variant gene cell) have a high similarity score 0.56 that suggests there may be a relationship between the variant gene cell and schizophrenia, which has been reported in several studies. Similarly, the high similarity score of 0.54 between topics 2 (pandemic anxiety health) and 8 (student university medication) indicates a potential relationship or overlap between these two topics. This finding suggests that there may be a connection between pandemic-related anxiety and issues related to students, universities, and medication. For example, the paper titled "Anxiety and Stress Levels Associated With COVID-

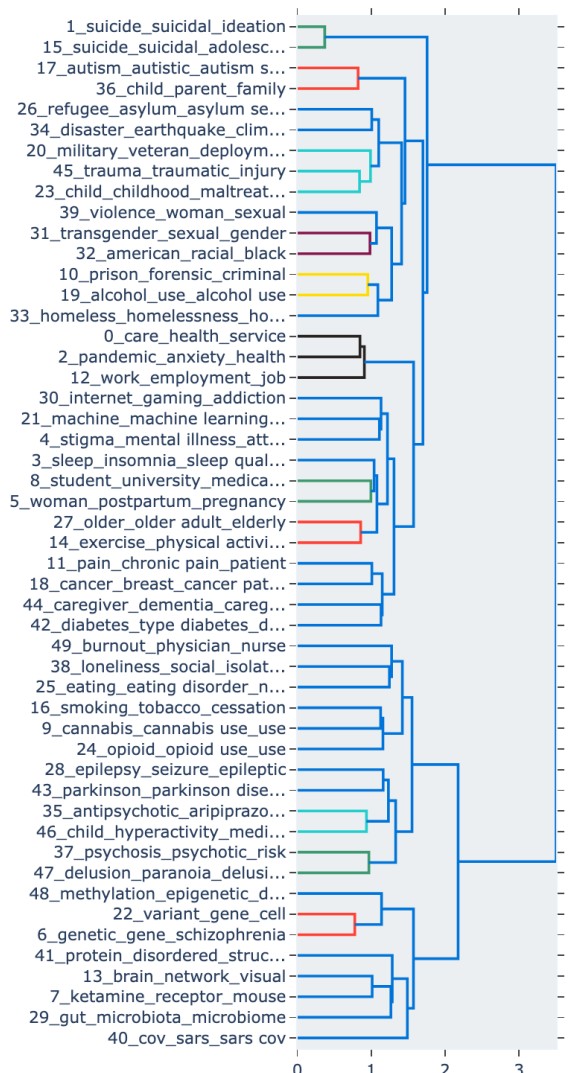

*Figure 4.* Hierarchical Clustering of Top 50 Most Frequent Mental Health Research Topics

19 Pandemic of University Students in Turkey: A Year After the Pandemic" published in the journal Frontiers in Psychiatry provides relevant information regarding the potential relationship between pandemic anxiety and university students. The research investigated the psychological impact of the pandemic on students and how it affected their mental health and well-being. The paper provides insights into the specific context of university students in Turkey and their experiences during the pandemic. It sheds light on the potential connection between pandemic anxiety and the well-being of students in a university setting. The paper could potentially contribute to the understanding of how pandemic-related anxiety impacts the mental health of university students and explore potential interventions, such as the use of medication, to mitigate these effects. By identi-

fying such relationships through the similarity matrix, researchers can gain a better understanding of the underlying structure of the field and the interconnections between different topics. This information can be valuable for various purposes, such as identifying important topics, exploring potential research directions, or discovering novel connections between seemingly unrelated topics.

Figure 4 depicts the hierarchical clustering of the Top 50 mental health-related topics. Hierarchical clustering allows topics to be organized into hierarchical structures or clusters based on their similarity, allowing for a more nuanced exploration of topic relationships. It can also help in summarizing large sets of topics by identifying representative topics within each cluster. For instance, topics 1 and 15 are found to be nearly identical, sharing similar content and underlying themes. Similarly, topics 17 and 36 revolve around the child-parent relationship and its impact on autism. The high similarity and hierarchical relationship between these two topics highlights the significant influence that the child-parent relationship plays a crucial role in the manifestation of autism. Furthermore, topics 9, 16, and 24 focus on distinct substances, namely tobacco, cannabis, and opioid use, respectively. These topics fall under the broader category of substance abuse.

In Figure 5, we harnessed the power of the gpt-3.5-turbo model from OpenAI API by providing it with a designed prompt. The prompt guided the API to assist us in identifying and filtering the primary machine learning methods employed in these research papers, as shown in Table 2. We further processed the data to generate a WordCloud figure. This figure visually represents the prevalence and prominence of different machine learning models within the field of mental health research across the specified time period. Each model is represented by a word, with the size of the word indicating its frequency and significance in the literature.

Our analysis revealed notable patterns in the usage of machine learning models over time. We observed a significant increase in the application of machine learning techniques as the years progressed. Additionally, prior to 2017, traditional machine learning methods were predominantly employed. However, after 2017, there was a remarkable surge in the utilization of neural networks, accompanied by the incorporation of natural language processing (NLP), computer vision (CV), and reinforcement learning (RL) algorithms. The WordCloud figure serves as a visual representation of these trends, offering a comprehensive snapshot of the evolving landscape of machine learning models in mental health research.

To evaluate the performance of the different topic models, we used a set of comprehensive metrics that assess topic coherence, diversity, and clustering quality. One of the

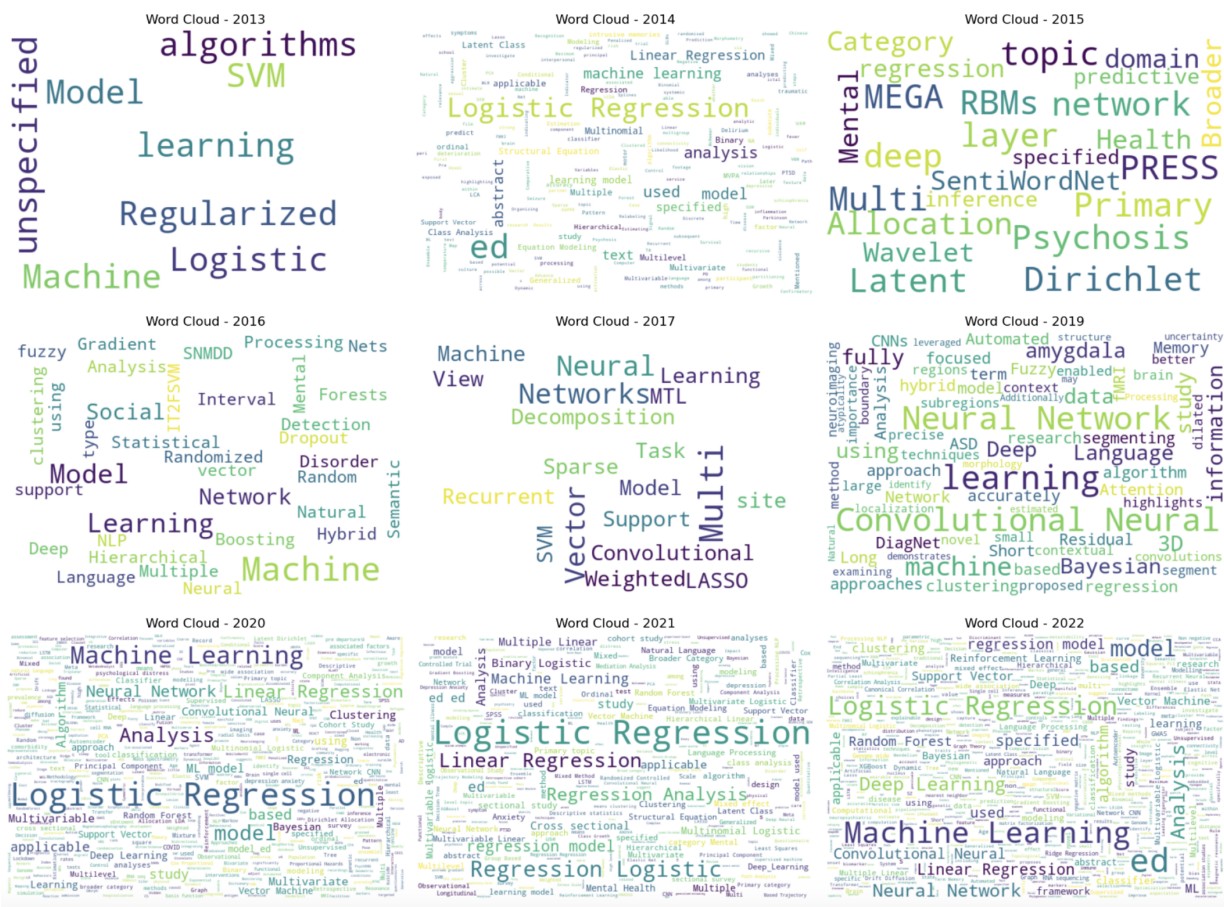

*Figure 5.* Temporal Evolution of Machine Learning Techniques in Mental Health Research: A Comparative WordCloud Analysis Across Different Years (2013-2017, 2019-2022)

metrics we employed was the NPMI (Normalized Pointwise Mutual Information), which measures the coherence of topic words by evaluating their association strength. A higher NPMI score, ranging from -1 to 1, indicates a stronger association among the topic words. We also utilized the Cv (Coefficient of Topic Coherence) measure, which calculates coherence using a larger sliding window and indirect cosine similarity. This measure has shown a high correlation with human judgment and ranges from 0 to 1, with 1 representing perfect association.

In addition to coherence, we measured the diversity of the topics using two metrics. The first metric, TD (Topic Diversity), calculates the percentage of unique topic words, with a score of 1 indicating that all topic words are different. The second metric, Inv. RBO (Inverted Rank-Biased Overlap), also measures topic diversity. It is a rank-weighted measure that gives less weight to words at higher ranks. Again, this metric ranges from 0 to 1, with 1 indicating all different topic words.

We presented the evaluation results of the BERTopic, LDA-BERT, and Top2Vec algorithms in Table 3. The results show that BERTopic outperformed the other algorithms in terms of TD and Cv scores. It achieved a significantly higher TD score of 0.7208 and Topic Coherence scores of 0.2892 (NPMI) and 0.8068 (Cv). Interestingly, Top2Vec performed well in terms of Inv. RBO with a score of 0.9669, although there is a smaller gap between Top2Vec and BERTopic in this metric.

## 6. Conclusion and Future Work

In conclusion, our study analyzed a large dataset of mental health research papers to identify trends and high-impact research topics. The heightened interest in mental health, particularly in response to the COVID-19 pandemic, has led to a surge in publications dedicated to this field. By leveraging a Sentence-BERT based embedding model and employing topic modeling techniques, we successfully examined topic evolution. The model demonstrated superior performance in diverse topic modeling metrics, indicating its effectiveness in generating distinct, coherent, and inter-

pretable topics. Additionally, by leveraging OpenAI GPT API, we extracted and analyzed machine learning models mentioned in the research papers, revealing computational techniques prevalent in mental health research. The resulting word cloud offered a comprehensive overview of the commonly utilized techniques and emerging trends. Overall, our findings contribute to understanding the landscape of mental health research landscape and provide insights for researchers and practitioners seeking to address mental health challenges using computational techniques.

We plan to expand beyond research papers to include diverse data sources like clinical records, social media data, and patient narratives in our future work. Incorporating these sources can offer a holistic view of mental health revealing new research directions and intervention strategies.

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

# A. Appendix

Figure 6 presents an intriguing analysis of the changing landscape of mental health topics over a span of twelve years. Through the utilization of WordCloud visualizations, this study aims to capture the key themes and trends surrounding mental health discussions across multiple years, spanning from 2011 to 2022. By comparing WordClouds from different time periods, we gain valuable insights into the evolving nature of public discourse surrounding mental health, shedding light on the shifting focus and priorities within this critical field. This figure serves as a visual representation, enabling researchers and practitioners to observe and explore the dynamic nature of mental health concerns, ultimately facilitating a deeper understanding of the societal factors that shape our attitudes and responses to mental well-being.

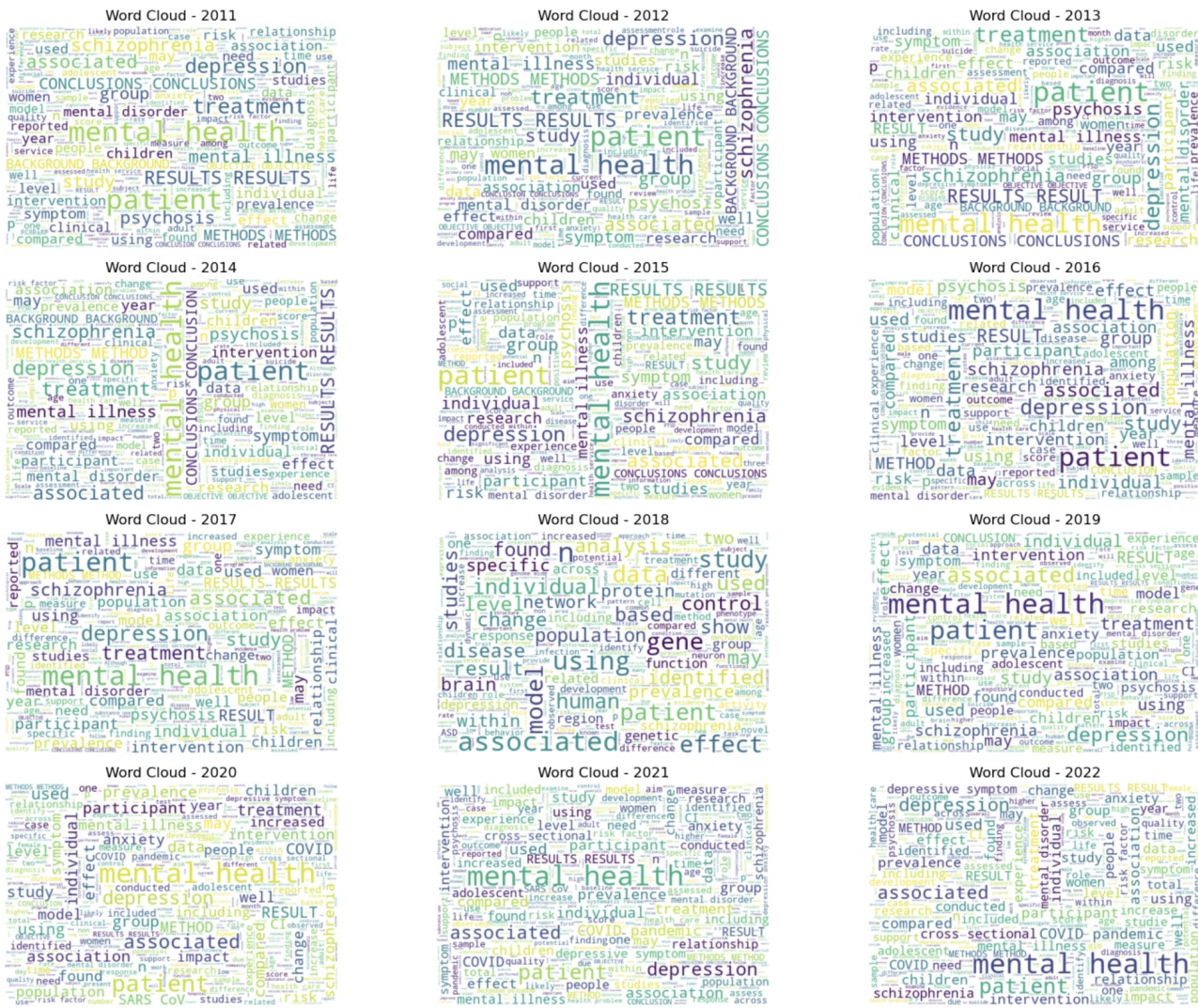

*Figure 6.* Temporal Evolution of Mental Health Topics: A Comparative WordCloud Analysis Across Different Years (2011-2022)

