# OpenReview forum: "Discovering Mental Health Research Topics with Topic Modeling"
_ICML.cc/2023/Workshop/IMLH — IMLH 2023 Poster_

### Official Review · Reviewer_Q9Cx · 2023-06-11
**Discovering Mental Health Research Topics with Topic Modeling**

**Rating:** 7
**Confidence:** 4

**Review:**

In this paper, the authors propose the use of Sentence-BERT, UMAP, HDBSCAN, and improved TF-IDF models to implement topic mining in mental health research. The authors have conducted experiments on 96,676 research papers collected from websites such as arXiv, ACM, and PubMed, and have displayed the changes in research topics in different years through word clouds. This study has important research significance for examining the changes and correlations of mental health research topics in different timestamps. At the same time, the paper gives a detailed experimental description and result analysis. However, the following content in the paper is still confusing.

Firstly, when the authors collect research paper data, they use AND/OR to connect keywords such as Agoraphobia, Anxiety Disorder, Attention-Deficit/Hyperactivity Disorder [ADHD], Autism Spectrum Disorder [ASD], and Post-Traumatic Stress Disorder [PTSD]. But, does this mean that there are artificially predefined subject words in the data collection stage? How did the authors preprocess the collected data and how did the authors define the boundary between model clustering results and subject terms?

Secondly, the author only compares with Top2Vec, LDA-BERT when evaluating the model's TD Unique, TC NPMI and other metrices. The empirical comparison experiment is too single and not convincing. In recent years, there have been many outputs of topic mining models, such as references 1 and 2.
1. Wang J Y, Zhang X L. Deep NMF topic modeling[J]. Neurocomputing, 2023, 515: 157-173.
2. Abadah M S K, Keikhosrokiani P, Zhao X. Analytics of Public Reactions to the COVID-19 Vaccine on Twitter Using Sentiment Analysis and Topic Modeling[M]//Handbook of Research on Applied Artificial Intelligence and Robotics for Government Processes. IGI Global, 2023: 156-188.

Finally, the experimental results did not well reflect the coherence of the topics discussed in the paper, so it was difficult to read along with the author's research ideas in the experimental results section, and the same things also happened in the method section.

---

### Official Review · Reviewer_UuzN · 2023-06-14
**A pioneer work on using large language model on an interesting topic**

**Rating:** 6
**Confidence:** 1

**Review:**

This paper focuses on discovering mental health research topics with large language model. Specifically, this paper collected about 96k papers, and use a large language model, BERTopic, to analysis those papers. In the experiments, compared to the other two methods, LDA-BERT and Top2Vec, BERTopic algorithm shows the best results.

This paper is a pioneer work on how to use the power of large models. I like the idea of this paper, which can be potentially used for other topic beyond mental health.

I also appreciate the open source effort, which could potentially help for further research efforts.

It would be better if there are more insightful understanding of how to choose models, and more understanding of the results.

---

### Official Review · Reviewer_ArT6 · 2023-06-17
**Review from Reviewer ArT6**

**Rating:** 5
**Confidence:** 4

**Review:**

The paper studies the topic discovering problem, which is quite relevant to the IMLH workshop, and helpful for interpretable ML. The only concern is the best model BERTopic is directly from an existing work. However, the authors claim it as "our model" in the draft. To me, this paper can be positioned as a paper with contributions in collecting datasets and applying existing models in domain-specific applications. I would suggest authors make this part cristal clear, and reduce the overclaim.

**Strength**
- Authors collect a large number of papers in the field of mental health from 2010 until the present.
- Great visualization. As a reader, these visualizations explain many findings and verified some claims in the draft.

**Weaknesses**
- The anonymous link shown in the draft is not accessible. I would suggest authors create such links using some popular tools: https://anonymous.4open.science/, https://figshare.com/. They all provide functions to enable anonymous links to existing codebase/database.
- The best model used in this paper, BERTopic, is an existing method from 2022. I would suggest authors avoid claiming BERTopic as "our model", as no signification improvement or modification is made.

---

### Meta-Review · Area_Chair_muiq · 2023-06-19

**Recommendation:** Accept (Poster)
**Confidence:** 3

**Metareview:**

This paper proposed approach for implementing topic mining in mental health research involves the use of Sentence-BERT, UMAP, HDBSCAN, and improved TF-IDF models. The authors have conducted experiments on a large dataset of 96,676 research papers collected from reputable websites such as arXiv, ACM, and PubMed. The results of the experiments are presented through word clouds, which showcase changes in research topics over different years.

The reviewers found the paper to be relevant to the workshop, well-motivated, and having implications for the interpretability field. Some concerns were raised regarding the need for improvement or modification, which should be taken into account for the next revision.

---

### Decision · Program_Chairs · 2023-06-20

Accept (Poster)